# Deep Patch Visual Odometry

**Zachary Teed**[*]
Princeton University
zteed@princeton.edu

**Lahav Lipson**[*]
Princeton University
llipson@princeton.edu

**Jia Deng**
Princeton University
jiadeng@princeton.edu

## Abstract

We propose Deep Patch Visual Odometry (DPVO), a new deep learning system for monocular Visual Odometry (VO). DPVO uses a novel recurrent network architecture designed for tracking image patches across time. Recent approaches to VO have significantly improved the state-of-the-art accuracy by using deep networks to predict dense flow between video frames. However, using dense flow incurs a large computational cost, making these previous methods impractical for many use cases. Despite this, it has been assumed that dense flow is important as it provides additional redundancy against incorrect matches. DPVO disproves this assumption, showing that it is possible to get the best accuracy and efficiency by exploiting the advantages of sparse patch-based matching over dense flow. DPVO introduces a novel recurrent update operator for patch based correspondence coupled with differentiable bundle adjustment. On Standard benchmarks, DPVO outperforms all prior work, including the learning-based state-of-the-art VO-system (DROID) using a third of the memory while running 3x faster on average. Code is available at https://github.com/princeton-vl/DPVO

## 1 Introduction

Visual Odometry (VO) is the task of estimating a robot's position and orientation from visual measurements. In this work, we focus on most challenging case—monocular VO—where the only input is a monocular video stream. The goal of the system is to estimate the 6-DOF pose of the camera at every frame while simultaneously building a map of the environment.

VO is closely related to Simultaneous Localization and Mapping (SLAM). Like VO, SLAM systems aim to estimate camera pose and map the environment but also incorporate techniques for global corrections—such as loop closure and relocalization (3). SLAM systems typically include a VO frontend which tracks incoming frames and performs local optimization.

Prior work typically treats VO as an optimization problem solving for a 3D model of the scene which best explains the visual measurements (3). *Indirect* approaches first detect and match keypoints between frames, then solve for poses and 3D points which minimize the reprojection distance (26; 4; 20). *Direct* approaches, on the other hand, operate directly on pixel intensities, attempting to solve for poses and depths which align the images (13; 15; 12). The main issue with prior systems, both direct and indirect, is the lack of robustness. Failure cases are too frequent for many important applications such as autonomous vehicles. These failure cases typically stem from moving objects, lost feature tracks, and poor convergence.

Several deep learning approaches (37; 45; 34; 39; 5; 43) have been introduced to address the robustness issue. The main advantage of deep learning is better features as well as differentiable optimization layers guided by neural networks. DROID-SLAM (37), which includes a VO frontend, uses neural networks to estimate dense flow fields which are subsequently used to optimize depth

---

[*]Equal Contribution

37th Conference on Neural Information Processing Systems (NeurIPS 2023).

and camera pose. DROID-SLAM has significantly improved the state-of-the-art in visual SLAM and VO. However, it comes with a large computational cost: in VO mode (without the backend), it averages 40FPS on an RTX-3090 using 8.7GB GPU memory, which can be impractical for resource-constrained devices. In this work, we aim to significantly improve the efficiency of deep VO, and therefore SLAM, without sacrificing its accuracy.

We propose Deep Patch Visual Odometry (DPVO), a new deep learning system for monocular VO. Our method achieves the accuracy and robustness of deep learning-based approaches while running 1.5-8.9x faster using 57%-29% the memory. It has lower average error than all prior work on common benchmarks. On an RTX-3090, it averages 60FPS using only 4.9GB of memory, with a minimum frame rate of 48FPS. It can run at 120FPS on the EuRoC dataset (2) using 2.5GB while still outperforming prior work on average. The frame rate of DPVO is relatively-constant in practice, and does not significantly depend on the degree of camera motion, as opposed to prior methods including DROID-SLAM which slow down during fast camera motion (12; 37; 27). We train our entire system end-to-end on synthetic data but demonstrate strong generalization on real video.

This leap in efficiency is achieved by the combination of **1)** a deep feature-based patch representation for keypoints which encodes their local context, and **2)** a novel recurrent architecture designed to track a sparse collection of these patches through time—alternating patch trajectory updates with a differentiable bundle adjustment layer to allow for end-to-end learning of reliable feature matching. This efficiency boost enables the design of DPVO to allocate resources towards components which improve accuracy, which are described in Sec. 3.

Existing approaches such as DROID-SLAM estimate camera poses by predicting dense flow (24; 23; 37). Designing an efficient VO system with the same-or-better accuracy poses a significant challenge, since using dense matches in SLAM provides additional redundancy against incorrect matches. However, DPVO still achieves robustness by introducing components which improve matching accuracy, using resources which would otherwise be spent estimating dense flow. We observe, surprisingly, that patch-based correspondence improves both efficiency and robustness over dense flow. In classical approaches to VO, patch-based matching (12) has been shown to improve accuracy over keypoint-based methods (27). However, it has remained unclear how to leverage this idea using deep networks without underperforming dense-flow approaches.

To summarize, we contribute Deep Patch Visual Odometry, a new method for tracking camera motion from video. DPVO uses a novel recurrent network designed for sparse patch-based correspondence. DPVO outperforms all prior work across several evaluation datasets, while running 1.5-8.9x faster than the previous state-of-the-art and using only 57-29% as much memory.

## 2   Related Work

Visual Odometry (VO) systems aim to estimate robot state (position and orientation) from a video. Overtime, a VO system will accumulate drift, and modern SLAM methods incorporate techniques to identify previously mapped landmarks to correct drift (i.e loop closure). VO can be considered a subproblem of SLAM with global optimization and loop closures disabled (3).

Many different modalities of VO have been explored by past work, including visual-inertial odometry (VIO) (41; 14) and stereo VO (42; 13). Here, we focus on the monocular case, where the only input is a monocular video stream. Early works approached the problem using filtering and maximum-likelihood methods (6; 25). Modern methods almost universally perform Maximum a Posteriori (MAP) estimation over factor graphs with Gaussian noise; in which case, the MAP estimate can be found by solving a non-linear least-squares optimization problem (8). This problem has lead to the development of many libraries for optimizing non-linear least-squares problems (1; 17; 7).

Among VO systems, our method borrows many core ideas of Direct Sparse Odometry (DSO) (12), a classical system based on least-squares optimization. Namely, we adopt a similar patch representation and reproject patches between frames to construct the objective function. Unlike DSO, the residuals are not based on intensity differences but instead predicted by a neural network which can pass information between patches and across patch lifetimes. Outlier rejection is automatically handled by the network, making our system more robust than classical systems like DSO and ORB-SLAM (12; 26). One important component of classical systems is the careful selection of

which image regions to use. We find, surprisingly, that our system works well on a small number (64 per frame) of *randomly* sampled image patches.

With regards to deep SLAM systems, our method is closely related to DROID-SLAM (37) but uses a different underlying representation and different network architecture. DROID-SLAM is an end-to-end deep SLAM system which shows good performance compared to both classical and deep baselines. Like our method, it works by iterating between motion updates and bundle adjustment. However, it estimates dense motion fields between selected pairs of frames which has a high computational cost and large memory footprint. While it is capable of 40FPS inference *on average*, its speed varies depending on the amount of motion in the video. Our method selects sparse patches from the video stream, with a relatively constant runtime per frame and 1.5-8.9x faster inference than DROID-SLAM.

Prior works such as BA-Net (35) and (21) also have embedded bundle adjustment layers in end-to-end differentiable network architectures. However, BA-Net does not use patch-based correspondence. (21) and (11) have proposed neural networks which sit atop COLMAP (31) and perform subpixel-level refinement; these approaches are not, however, able to perform 3D reconstruction on their own and are subject to failure cases in the underlying SfM system.

## 3  Approach

**Preliminaries:**  Given an input video, we represent a scene as a collection of camera poses $\mathbf{T} \in \mathbb{SE}(3)^N$ and a set of square image patches $\mathbf{P}$ extracted from the video frames. Using $\mathbf{d}$ to represent inverse depth and $(\mathbf{x}, \mathbf{y})$ to represent pixel coordinates, we represent each patch as the $4 \times p^2$ homogeneous array

$$\mathbf{P}_k = \begin{pmatrix} \mathbf{x} \\ \mathbf{y} \\ 1 \\ \mathbf{d} \end{pmatrix} \qquad \mathbf{x}, \mathbf{y}, \mathbf{d} \in \mathbb{R}^{1 \times p^2} \tag{1}$$

where $p$ is the width of the patch. We assume a constant depth for the full patch, meaning that it forms a fronto-parallel plane in the frame from which it was extracted. Letting $i$ denote the index of the *source frame* of the patch, i.e. the frame from which which patch $\mathbf{P}_k$ was extracted, we can reproject the patch onto another frame $j$

$$\mathbf{P}'_{kj} \sim \mathbf{K}\mathbf{T}_j\mathbf{T}_i^{-1}\mathbf{K}^{-1}\mathbf{P}_k. \tag{2}$$

taking $\mathbf{K}$ to be the $4 \times 4$ calibration matrix

$$\mathbf{K} = \begin{pmatrix} f_x & 0 & c_x & 0 \\ 0 & f_y & c_y & 0 \\ 0 & 0 & 1 & 0 \\ 0 & 0 & 0 & 1 \end{pmatrix} \tag{3}$$

The pixel coordinates $\mathbf{x}' = (x', y')$ can be recovered by dividing by the third element. For the rest of the paper, we use the shorthand $\mathbf{P}'_{kj} = \omega_{ij}(\mathbf{T}, \mathbf{P}_k)$ to denote the reprojection of patch $k$ onto frame $j$ in terms of pixel coordinates.

*Patch Graph:* We use a bipartite *patch graph* to represent the relations between patches and video frames. Edges in the graph connect patches with frames. We show an example in Fig. 1. By default, the graph is constructed by adding an edge between each patch and every frame within distance $r$ from the index of the source frame of the patch. The reprojections of a patch in all of its connected frames in the patch graph form the *trajectory* of the patch. Note that a trajectory is a set of reprojections of a single patch into multiple frames; it is not a set of original square patches. The graph is dynamic; as new video frames are received, new frames and patches are added while old ones are removed. We provide an example trajectory in Fig. 3.

**Approach Overview:**  At a high level, our approach works similarly to a classical system: it samples a set of patches for each video frame, estimates the 2D motion (optical flow) of each patch against each of its connected frames in patch graph, and solves for depth and camera poses that are consistent with the 2D motions. But our approach differs from a classical system in that these steps are done through a recurrent neural network and a differentiable optimization layer. Our approach can be understood as a sparsified variant of DROID-SLAM in VO mode; instead of estimating dense optical flow, we estimate flow for a sparse set of patches.

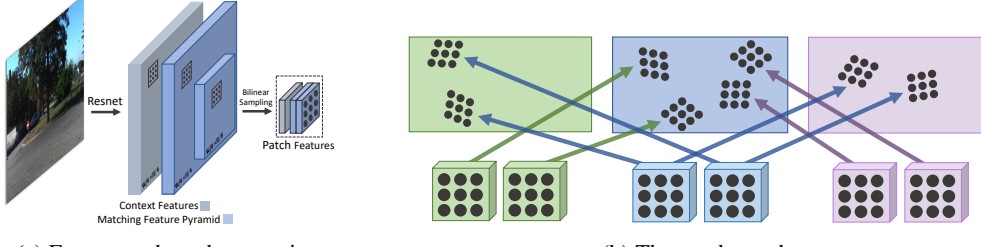

(a) Feature and patch extraction                (b) The patch graph

Figure 1: Constructing the patch graph. (a) Residual networks extract 1) a context feature map at $\frac{1}{4}$ image-resolution and 2) a 2-level pyramid of matching features at $\frac{1}{4}$ and $\frac{1}{16}$ resolution. Many $p \times p$ patches are cropped from this feature map at *random* pixel coordinates using bilinear sampling. (b) Multiple patches are extracted from each frame (e.g. blue) and are connected to nearby frames (green and purple). In the resulting patch graph, edges connect patches with frames.

**Feature Extraction:** To estimate the optical flow of the patches, we need to extract per-pixel features and use them for computing visual similarities. We use a pair of residual networks. One network extracts *matching* features while the other extracts *context* features. The first layer of each network is a $7 \times 7$ convolution with stride 2 followed by two residual blocks at 1/2 resolution (dimension 32) and 2 residual blocks at 1/4 resolution (dimension 64), such that the final feature map is one-quarter the input resolution. The architectures of the matching and context networks are identical with the exception that the matching network uses instance normalization and the context network uses no normalization. We construct a two-level feature pyramid by applying average pooling to the matching features with a $4 \times 4$ filter with stride 4.

**Patch Extraction:** We create patches by randomly sampling 2D locations, which we find to work well despite its simplicity. We associate each patch with a per-pixel feature map of the same size, cropped with bilinear interpolation from both the full matching and full context feature maps. We provide an example in Fig. 1a. The patch features are used to compute visual similarities, but unlike DROID-SLAM, we compute them on the fly instead of precomputing them as correlation volumes.

### 3.1 Update Operator

The core of our approach is the update operator, which is a recurrent network that iteratively refines the depth of each patch and the camera pose of each video frame. At each iteration, it uses the patch features to propose revisions to the optical flow of the patches, and updates depth and camera poses through a differentiable bundle adjustment layer. Because the patches are sparsely located and spatially separated, the recurrent network includes special designs that facilitate exchange of information between patches. We provide a schematic overview of the operator in Fig. 2. The operator operates on the patch graph and maintains a hidden state for each edge. Its first three components (Correlation, Temporal Convolutions, Softmax-Aggregation) produce and aggregate information across edges, the Transition block produces an update to each hidden state, and the final two components (Factor-Head + Bundle-Adjustment) produce an update to the camera poses and patch depths. When a new edge is added, its hidden state is initialized with zeros.

*Correlation*: For each edge $(k, j)$ in the patch graph, we compute correlation features (visual similarities) to assess the visual alignment given by current estimates of depth and poses and to propose revisions. We first use Eqn. 2 to reproject patch $k$ from frame $i$ into frame $j$: $\mathbf{P}'_{kj} = \omega_{ij}(\mathbf{T}, \mathbf{P}_k)$. Given patch features $\mathbf{g} \in \mathbb{R}^{p \times p \times D}$ and frame features $\mathbf{f} \in \mathbb{R}^{H \times W \times D}$, for each pixel $(u, v)$ in patch $k$, we compute its correlation $\mathbf{C} \in \mathbb{R}^{p \times p \times 7 \times 7}$ with a grid of pixels centered at its reprojection in frame $j$, using the inner product:

$$\mathbf{C}_{uv\alpha\beta} = \langle \mathbf{g}_{uv}, \ \mathbf{f}(\mathbf{P}'_{kj}(u, v) + \Delta_{\alpha\beta}) \rangle \tag{4}$$

where we take $\Delta$ to be a $7 \times 7$ integer grid centered at 0 indexed by $\alpha$ and $\beta$, and $\mathbf{f}(\cdot)$ denotes bilinear sampling. We compute these correlation features for both levels in the pyramid and concatenate the results. This operation is implemented as an optimized CUDA layer which leverages the regular grid structure of the interpolation step. This implementation is identical to the alternative correlation

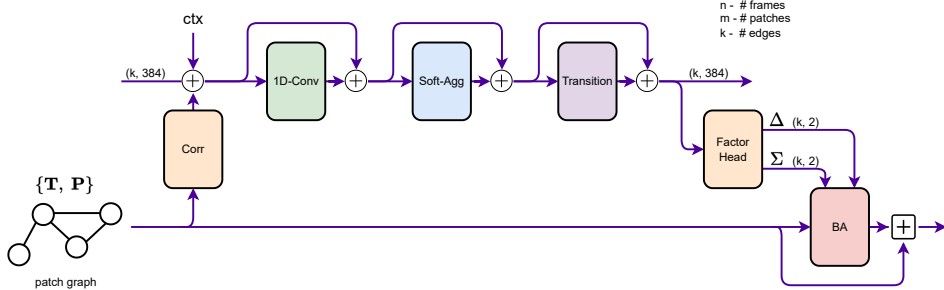

Figure 2: Schematic of the *update operator*. Correlation features are extracted from edges in the patch graph and injected into the hidden state alongside context features. We apply 1D convolution, message passing and a transition block. The factor head produces trajectory revisions which are used by the bundle adjustment layer to update the camera poses and the depth of patches. Each "+" operation is a residual connection followed by layer normalization.

implementation used by RAFT (36) and is equivalent to indexing correlation volumes due to the linearity of the inner product and interpolation.

***1D Temporal Convolution****:* DPVO operates on real-time video, where neighboring frames tend to be highly correlated. To leverage this correlation, we apply a 1D-convolution in the temporal dimension to each patch trajectory. Since trajectories vary in length and keyframes are actively added and removed, it is not straightforward to implement convolution as a batched operation. Instead, for each edge $(k, j)$ we index the features of its temporally-adjacent neighbors at $(k, j - 1)$ and $(k, j + 1)$, concatenate, then apply a linear projection. The temporal convolution allows the network to propagate information along each patch trajectory and model appearance changes of the patch through time.

***SoftMax Aggregation****:* Even though the patches are sparsely located, their motion and appearance can be still be correlated as they may belong to the same object. We leverage this correlation through global message passing layers that propagate information between edges in the patch graph. This operation has appeared before in the context of graph neural networks (30). Given edge $e$ and its neighbors $N(e)$ we define the channel-wise aggregation function

$$\psi \left( \left[ \sum_{x \in N(e)} \sigma(x) \cdot \phi(x) \right] \Big/ \sum_{x \in N(e)} \sigma(x) \right) \tag{5}$$

where $\psi$ and $\phi$ are linear layers and $\sigma$ is a linear layer followed by a sigmoid activation. We perform two instantiations of soft aggregation: (1) patch aggregation where edges are neighbors if they connect to the same patch (2) frame aggregation where edges are neighbors if they connect to both the same destination frame and different patches from the same source frame. Note that this graph-based aggregation is unique to our sparse patch-based representation, because for dense flow the same effect is easily achieved with convolution in prior approaches (37).

***Transition Block****:* Following the softmax-aggregation, we include a *transition block* (shown in Fig. 2) to produce an update to the hidden state for each edge in the patch graph. Our transition block is simply two gated residual units with Layer Normalization and ReLU non-linearities. We find that layer normalization helps avoid exploding values for recurrent network modules.

***Factor Head****:* The final *learned* layer in our update operator is the *factor head*, which proposes 2D revisions to the current patch trajectories and associated confidence weights. This layer consists of 2 MLPs with one hidden unit each. For each edge $(k, j)$ in the patch graph, the first MLP predicts the trajectory update $\delta_{kj} \in \mathbb{R}^2$: a 2D flow vector indicating how the reprojection of the patch center should be updated in 2D; the second MLP predicts the confidence weight $\Sigma_{kj} \in \mathbb{R}^2$ which is bounded to $(0, 1)$ using a sigmoid activation.

***Differentiable Bundle Adjustment****:* Given the proposed 2D revisions to trajectories, we need to solve for the updates to depth and camera poses to realize the proposed revisions. This is achieved through a differentiable bundle adjustment (BA) layer, which operates globally on the patch graph and outputs updates to depth and camera poses. The predicted factors $(\delta, \Sigma)$ are used to define an

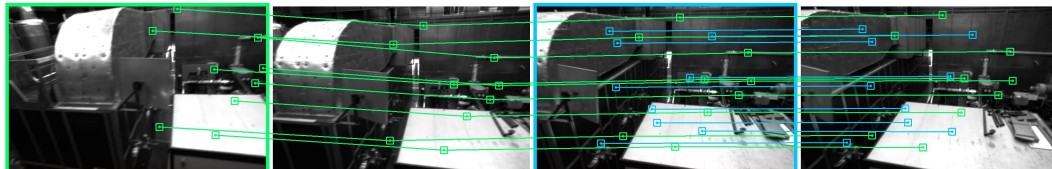

Figure 3: A subset of the patch trajectories predicted by our method. Patches extracted from the green keyframe are tracked through subsequent frames. When a new keyframe is added (blue), additional patches are extracted and tracked. Our method produces confidence values which weight their respective contribution to the bundle adjustment.

optimization objective:

$$\sum_{(k,j)\in\mathcal{E}} \left\| \hat{\omega}_{ij}(\mathbf{T}, \mathbf{P}_k) - [\hat{\mathbf{P}}'_{kj} + \delta_{kj}] \right\|^2_{\Sigma_{kj}} \tag{6}$$

where $\|\cdot\|_\Sigma$ is the Mahalanobis distance and $\hat{\mathbf{P}}'_{kj}$ denotes the center of $\mathbf{P}'_{kj}$. We apply two Gauss-Newton iterations to the linearized objective, optimizing the camera poses as well as the inverse depth component of the patch while keeping the pixel coordinates constant. This optimization seeks to refine the camera poses and depth such that the induced trajectory updates agree with the predicted trajectory updates. Like DROID-SLAM (37), we use the Schur complement trick for efficient decomposition and backpropagate gradients through the Gauss-Newton iterations.

### 3.2 Training and Supervision

DPVO is implemented using PyTorch. We perform supervised training of our network on the TartanAir dataset. On each training sequence, we precompute optical flow magnitude between all pairs of frames using ground truth poses and depth. During training, we sample trajectories where frame-to-frame optical flow magnitude is between 16px and 72px. This ensures that training instances are generally difficult but not impossible.

We apply supervision to poses and induced optical flow (i.e. trajectory updates), supervising each intermediate output of the update operator and detach the poses and patches from the gradient tape prior to each update.

***Pose Supervision:*** We scale the predicted trajectory to match the groundtruth using the Umeyama alignment algorithm (40). Then for every pair of poses $(i, j)$, we supervise on the error

$$\sum_{(i,j)\ i\neq j} \| Log_{SE(3)}[(\mathbf{G}_i^{-1}\mathbf{G}_j)^{-1}(\mathbf{T}_i^{-1}\mathbf{T}_j)]\| \tag{7}$$

where $\mathbf{G}$ is the ground truth and $\mathbf{T}$ are the predicted poses.

***Flow Supervision:*** We additionally supervise on the distance between the induced optical flow and the ground truth optical flow between each patch and the frames within two timestamps of its source frame. Each patch induces a $p \times p$ flow field. We take the minimum of all $p \times p$ errors.

The final loss is the weighted combination

$$\mathcal{L} = 10\mathcal{L}_{pose} + 0.1\mathcal{L}_{flow}. \tag{8}$$

***Training Details:*** We train for a total of 240k iterations on a single RTX-3090 GPU with a batch size of 1. Training takes 3.5 days. We use the AdamW optimizer and start with an initial learning rate of 8e-5 which is decayed linearly during training. We apply standard augmentation techniques such as resizing and color jitter.

We train on sequences of length 15. The first 8 frames are used for initialization while the next 7 frames are added one at a time. We unroll 18 iterations of the update operator during training. For the first 1000 training steps, we fix poses with the ground truth and only ask the network to estimate the depth of the patches. Afterwards, the network is required to estimate both poses and depth.

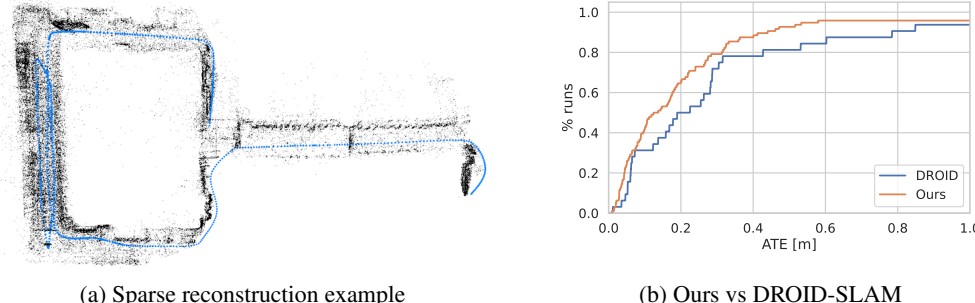

| (a) Sparse reconstruction example | (b) Ours vs DROID-SLAM |

Figure 4: Results on the TartanAir (44) validation split. Our method gets an AUC of 0.80 compared to 0.71 for DROID-SLAM while running 4x faster.

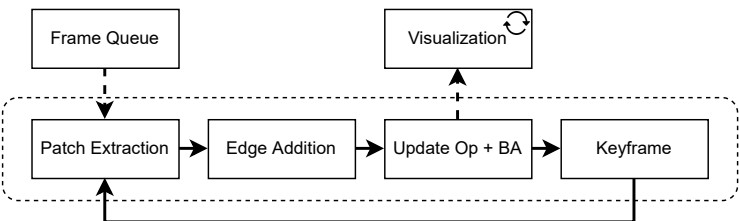

Figure 5: Overview of the VO System. The visualization and frame loading are performed in separate threads.

### 3.3 VO System

We implement the logic of the full VO system primarily in Python with bottleneck operations such as bundle adjustment and visualization implemented in C++ and CUDA.

*Overview:* In Fig. 5 we show an overview of the VO system. The visualization and frame loading are performed in separate threads.

*Initialization:* We use 8 frames for initialization. We add new patches and frames until 8 frames are accumulated and then run 12 iterations of our update operator. There needs to be some camera motion for initialization; hence, we only accumulate frames with an average flow magnitude of at least 8 pixels from the prior frame.

*Expansion:* When a new frame is added we extract features and patches. The pose of the new frame is initialized using a constant velocity motion model. The depth of the patch is initialized as the median depth of all the patches extracted from the previous 3 frames.

We connect each patch to every frame within distance $r$ from the frame index where the patch was extracted. This means that when a new patch is added, we add edges between that patch and the previous $r$ keyframes. When a new frame is added, we add edges between each patch extracted in the last $r$ keyframes with the new frame. This strategy means that each patch is connected to no more than $(2r - 1)$ frames, bounding the worst-case latency.

*Optimization:* Following the addition of edges we run one iteration of the update operator followed by two bundle adjustment iterations. We fix the poses of all but the last 10 keyframes. The inverse depths of all patches are free parameters. The patches are removed from optimization once they fall outside the optimization window.

*Keyframing:* The most recent 3 frames are always taken to be keyframes. After each update, we compute the optical flow magnitude between keyframe $t - 5$ and $t - 3$. If this is less than 64px, we remove the keyframe at $t - 4$. When a keyframe is removed, we store the relative pose between its neighbors such that the full pose trajectory can be recovered for evaluation.

| | ME 000 | ME 001 | ME 002 | ME 003 | ME 004 | ME 005 | ME 006 | ME 007 | MH 000 | MH 001 | MH 002 | MH 003 | MH 004 | MH 005 | MH 006 | MH 007 | Avg |
|---|---|---|---|---|---|---|---|---|---|---|---|---|---|---|---|---|---|
| ORB-SLAM3* (4) | 13.61 | 16.86 | 20.57 | 16.00 | 22.27 | 9.28 | 21.61 | 7.74 | 15.44 | 2.92 | 13.51 | 8.18 | 2.59 | 21.91 | 11.70 | 25.88 | 14.38 |
| COLMAP* (31) | 15.20 | 5.58 | 10.86 | 3.93 | 2.62 | 14.78 | 7.00 | 18.47 | 12.26 | 13.45 | 13.45 | 20.95 | 24.97 | 16.79 | 7.01 | 7.97 | 12.50 |
| DSO (12) | 9.65 | 3.84 | 12.20 | 8.17 | 9.27 | 2.94 | 8.15 | 5.43 | 9.92 | 0.35 | 7.96 | 3.46 | - | 12.58 | 8.42 | 7.50 | 7.32 |
| DROID-SLAM* (37) | 0.17 | **0.06** | 0.36 | 0.87 | 1.14 | **0.13** | 1.13 | **0.06** | **0.08** | 0.05 | **0.04** | **0.02** | **0.01** | 0.68 | 0.30 | **0.07** | 0.33 |
| DROID-VO | 0.22 | 0.15 | 0.24 | 1.27 | 1.04 | 0.14 | 1.32 | 0.77 | 0.32 | 0.13 | 0.08 | 0.09 | 1.52 | 0.69 | 0.39 | 0.97 | 0.58 |
| Ours (Default) | **0.16** | 0.11 | **0.11** | 0.66 | **0.31** | 0.14 | **0.30** | 0.13 | 0.21 | **0.04** | **0.04** | 0.08 | 0.58 | **0.17** | 0.11 | 0.15 | **0.21** |
| Ours (Fast) | 0.35 | 0.13 | 0.27 | 0.71 | 0.47 | 0.16 | **0.30** | 0.13 | 0.34 | 0.05 | 0.06 | 0.07 | 0.81 | 0.41 | **0.09** | 0.14 | 0.28 |

Table 1: Results on the TartanAir monocular test split from the ECCV 2020 SLAM competition. Results are reported as ATE with scale alignment. For our method, we report the median of 5 runs. Methods marked with (*) use global optimization / loop closure.

| | 360 | desk | desk2 | floor | plant | room | rpy | teddy | xyz | avg |
|---|---|---|---|---|---|---|---|---|---|---|
| ORB-SLAM3[4] | x | **0.017** | 0.210 | x | 0.034 | x | x | x | **0.009** | - |
| DSO[13] | 0.173 | 0.567 | 0.916 | 0.080 | 0.121 | 0.379 | 0.058 | x | 0.036 | - |
| DSO-Realtime[13] | 0.172 | 0.718 | 0.728 | 0.068 | 0.167 | 0.767 | x | x | 0.031 | - |
| DROID-VO[32] | 0.161 | 0.028 | 0.099 | **0.033** | **0.028** | **0.327** | **0.028** | 0.169 | 0.013 | 0.098 |
| Ours (Default) | **0.135** | 0.038 | **0.048** | 0.040 | 0.036 | 0.394 | 0.034 | **0.064** | 0.012 | **0.089** |
| Ours (Fast) | 0.169 | 0.029 | 0.064 | 0.047 | 0.047 | 0.396 | 0.034 | 0.074 | 0.012 | 0.097 |

Table 2: Results (ATE) on the freiburg1 set of TUM-RGBD (33). This evaluates monocular visual odometry (No method uses stereo or sensor depth), and is identical to the evaluation setting in DROID-SLAM (37). For each sequence, we report the median result across 5 independent trials. Missing values (i.e. x) indicate that the method completely failed and refused to output a trajectory. Only learning based methods are robust enough to not fail on any sequence, and out of them Ours (Default) performs best on average.

## 4  Experiments

We evaluate DPVO on the TartanAir (44), TUM-RGBD (33), EuRoC (2) and ICL-NUIM (19) benchmarks. On each dataset, we run five trials each with a different set of patches and report the median results obtained. Example reconstructions on the TartanAir and ETH3D datasets are shown in Fig. 4a and in the Appendix, respectively. We benchmark two configuration settings of DPVO: Ours (Default) uses 96 patches per image and a 10 frame optimization window and Ours (Fast) uses 48 patches and a 7 frame optimization window. Our frame rates are practically near constant, since our method uses a constant number of FLOPS per-frame, unlike prior methods which slow down and make more keyframes during fast motion (37; 12). We also benchmark two versions of DROID-SLAM: the full version and a version where loop closure and global bundle adjustment is disabled (DROID-VO). DROID-VO is more comparable to our method as we only perform local optimization. Across all benchmarks, DPVO achieves the lowest average error among all previous VO systems, and even outperforms full SLAM systems on some benchmarks. We focus on average error as it reflects the expected performance of our system in the wild.

**TartanAir Validation Split:** We use the same 32-sequence validation split as DROID-SLAM and report aggregated results in Fig. 4b and compare with DROID-SLAM and ORB-SLAM3. We run our method 3 times on each sequence and aggregate the results. In the $[0, 1]$m error window, we get an AUC of 0.80 compared to 0.71 for DROID-SLAM.

**TartanAir Test Split:** Tab. 1 reports results on the test-split used in the ECCV 2020 SLAM competition compared to state-of-the-art methods. Classical methods such as DSO and ORB-SLAM fail on more than 80% of the sequences, hence we use COLMAP as a classical baseline as it was used in the two winning solutions of the ECCV SLAM competition. We attain an error 40% lower than DROID-SLAM and 64% lower than DROID-VO. COLMAP takes 2 days to complete the 16 sequences and typically produces broken reconstructions which lead to large errors in evaluation.

**EuRoC MAV (2)** In Tab. 3 we benchmark on the EuRoC MAV (2) dataset and compare to other visual odometry methods including SVO (15), DSO (12) and the visual odometry version of DROID-SLAM (36). Video from the EuRoC benchmark is recorded at 20FPS. Like DROID-SLAM, we skip every other frame, doubling the effective frame rate of the system. Our default system outperforms prior work on the majority of the EuRoC sequences. The average error is 43% lower than DROID-VO (37). Even our 120-FPS system outperforms DROID-VO on most video.

| | MH01 | MH02 | MH03 | MH04 | MH05 | V101 | V102 | V103 | V201 | V202 | V203 | Avg |
|---|---|---|---|---|---|---|---|---|---|---|---|---|
| TartanVO (43) | 0.639 | 0.325 | 0.550 | 1.153 | 1.021 | 0.447 | 0.389 | 0.622 | 0.433 | 0.749 | 1.152 | 0.680 |
| SVO (15) | 0.100 | 0.120 | 0.410 | 0.430 | 0.300 | 0.070 | 0.210 | - | 0.110 | 0.110 | 1.080 | 0.294 |
| DSO (12) | **0.046** | **0.046** | 0.172 | 3.810 | **0.110** | 0.089 | **0.107** | 0.903 | **0.044** | 0.132 | 1.152 | 0.601 |
| DROID-VO (37) | 0.163 | 0.121 | 0.242 | 0.399 | 0.270 | 0.103 | 0.165 | 0.158 | 0.102 | 0.115 | **0.204** | 0.186 |
| Ours (Default) | 0.087 | 0.055 | **0.158** | **0.137** | 0.114 | **0.050** | 0.140 | **0.086** | 0.057 | **0.049** | 0.211 | **0.105** |
| Ours (Fast) | 0.101 | 0.067 | 0.177 | 0.181 | 0.123 | 0.053 | 0.158 | 0.095 | 0.095 | 0.063 | 0.310 | 0.129 |

Table 3: Monocular SLAM on the EuRoC datasets, ATE[m] compared to other visual odometry methods. For our method, we report the median of 5 runs.

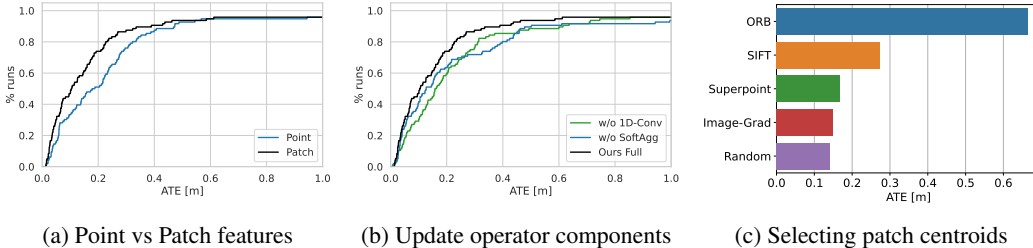

(a) Point vs Patch features          (b) Update operator components          (c) Selecting patch centroids

Figure 6: Ablation experiments. (a) We show the importance of using patches over point features. (b) Removal of different components of the update operator degrades accuracy. (c) Randomly selecting patch centroids works better than 2D keypoint locations produced via SIFT (22), ORB (29), Superpoint (9), or pixels with high image gradient.

**TUM-RGBD (33):** In Tab. 2 we benchmark on TUM-RGBD (33) and compare to DSO (12), the visual odometry version of DROID-SLAM (36), and ORB-SLAM3 (27). We evaluate only visual-only monocular methods, identical to the TUM-RGBD evaluation in (37). This benchmark evaluates motion tracking in an indoor environment with erratic camera motion and significant motion blur. Video from the TUM-RGBD benchmark is recorded at 30FPS.

Classical approaches such as ORB-SLAM (27) and DSO (12) perform well on some sequences but exhibit frequent catastrophic failures. Similar to DROID-VO, our system is also robust to such failures but our average error is 9% lower. This indicates that our sparse, patch-based approach performs better in-the-wild compared to dense flow.

**ICL-NUIM (19):** In Tab. 4 we evaluate on the ICL-NUIM (19) SLAM benchmark and compare to other visual odometry and SLAM methods including SVO (15), DSO (12) and DROID-SLAM (37). Video from the ICL-NUIM benchmark is recorded at 30FPS. ICL-NUIM is a synthetic dataset for evaluating SLAM performance in indoor environments with repetitive or monochrome textures (e.g. blank white walls). Our system outperforms prior work on the majority of the ICL-NUIM sequences.

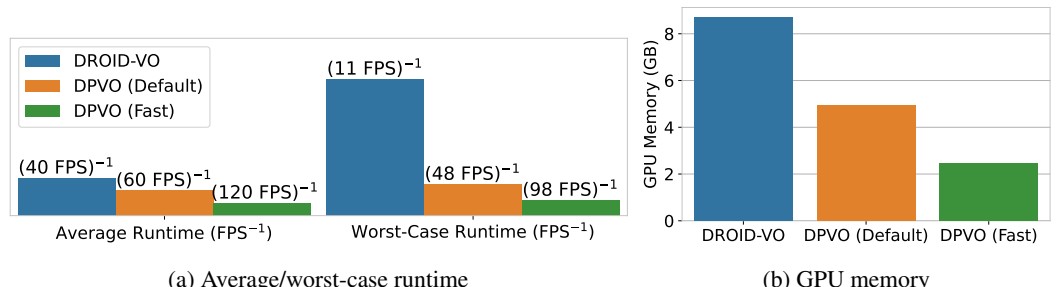

(a) Average/worst-case runtime          (b) GPU memory

Figure 7: Efficiency on EuRoC. (a) Compared to DROID-VO (the front-end of DROID-SLAM), the *Default* and *Fast* variants of DPVO run 1.5x (60FPS) and 3x (120FPS) faster on average. DPVO's frame-rate is relatively stable, remaining above 48FPS and 98FPS for 95% of frames. In contrast, DROID-VO drops to 11FPS, an 8.9x difference. (b) DPVO (Default) and DPVO (Fast) use 57% and 29% as much memory as DROID-VO, respectively. (c) We can trade-off speed for performance by increasing the number of patches tracked. However, performance quickly saturates beyond 96.

| | Living Room 0 | Living Room 1 | Living Room 2 | Living Room 3 | Office Room 0 | Office Room 1 | Office Room 2 | Office Room 3 | Avg |
|---|---|---|---|---|---|---|---|---|---|
| ORB-SLAM2* (27) | 0.461 | 0.172 | 0.806 | N/A | 0.589 | 0.813 | 1.000 | N/A | N/A |
| DROID-SLAM* (37) | 0.008 | 0.027 | 0.039 | 0.012 | **0.065** | 0.025 | 0.858 | 0.481 | 0.189 |
| DROID-VO (37) | 0.010 | 0.123 | 0.072 | 0.032 | 0.095 | 0.041 | 0.842 | 0.504 | 0.215 |
| SVO (15) | 0.02 | 0.07 | 0.09 | 0.07 | 0.34 | 0.28 | 0.14 | **0.08** | 0.136 |
| DSO (12) | 0.01 | 0.02 | 0.06 | 0.03 | 0.21 | 0.83 | 0.36 | 0.64 | 0.270 |
| DSO (Realtime) (12) | 0.02 | 0.03 | 0.33 | 0.06 | 0.29 | 0.64 | 0.23 | 0.46 | 0.258 |
| Ours (Default) | **0.006** | **0.006** | 0.023 | **0.010** | 0.067 | **0.012** | **0.017** | 0.635 | 0.097 |
| Ours (Fast) | 0.008 | 0.007 | **0.021** | **0.010** | 0.071 | 0.015 | 0.018 | 0.593 | **0.093** |

Table 4: Results on the ICL-NUIM SLAM benchmark. Results are reported as ATE with scale alignment. Methods marked with (*) use global optimization / loop closure. SVO and DSO results are from (16). ORB-SLAM2 results are from (10) (only partial results were provided). Despite our best efforts, we were unsuccessful in running monocular ORB-SLAM3: The official software would not produce any keyframes while running.

The average error of our faster model is $51\%$ lower than DROID-SLAM (37) and $32\%$ lower than SVO (15). Our faster and default systems perform similarly.

**Efficiency:** In Fig. 7 we compare the efficiency of DPVO and DROID-VO. Our 60-FPS variant is 1.5x faster on average and 4.3x faster in the worst case (5th percentile), using 57% the GPU memory of DROID-VO. The 120-FPS DPVO variant is 3x faster on average and 8.9x faster in the worst-case, using less than a third of the memory.

### 4.1 Ablations

We perform ablation experiments on the TartanAir validation split and show results in Fig. C. We use the same parameter settings in all experiments with augmentation disabled. We run each ablation experiment three times on the validation split and aggregate the results.

*Point vs Patch Features*: In Fig. 6a, we demonstrate the importance of the patches over simply using point features (i.e 1x1 patches). The patch features encode local context which is lacking with point features. The additional information stored in the correlation features allows for more precise tracking.

*Update Operator*: In Fig. 6b, we test the effect of removing various components from the update operator. Both removing 1D-Convolution and Softmax-Aggregation degrade performance on the validation set.

*Patch Selection*: In Tab. 6c, we compare different methods for selecting 2D patch centroids. Surprisingly, our method performs best when *randomly* selecting keypoint centroids.

## 5 Conclusion

DPVO is a new deep visual odometry system built using a sparse patch representation. It is accurate and efficient, capable of running at 60-120 FPS with minimal memory requirements. DPVO outperforms all prior work (classical or learned) on EuRoC, TUM-RGBD, the TartanAir ECCV 2020 SLAM competition, and ICL-NUIM.

**Acknowledgments:** This work was partially supported by the Princeton University Jacobus Fellowship, NSF, Qualcomm, and Amazon.

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

## A  Stable Runtime

The runtime of our approach is relatively constant compared to prior VO/SLAM systems. Our default configuration averages 60FPS, and rarely drops below 50FPS. In Fig. A, we show the runtime distribution on EuRoC. The runtime stability of DPVO results from the comparatively simple keyframing mechanism.

**Discussion on Keyframing:** Most VO/SLAM systems (12; 27; 37), including DROID-SLAM, contain a protocol for deciding when to produce new keyframes from a video stream. Keyframing mechanisms serve to discard repetitious frames early-on, saving on computation and potentially improving performance by limiting redundancy. Many VO/SLAM systems require a low keyframing frequency in order to run at real-time framerates. If there is significant motion between subsequent frames, keyframes are created at a high frequency which can cause such methods to slow down drastically. DROID-SLAM, for instance, slows from 40FPS to 11FPS during sufficiently large increases in the number of keyframes being created.

Unlike previous works, DPVO treats *all* incoming frames as keyframes and only removes redundant frames later using motion computed from the already-estimated pose. Although this design decision is sub-optimal (in terms of speed) during very slow or still camera movement (e.g. EuRoC), our approach is both simpler and ensures our frame-rate is approximately constant no matter the degree of camera motion.

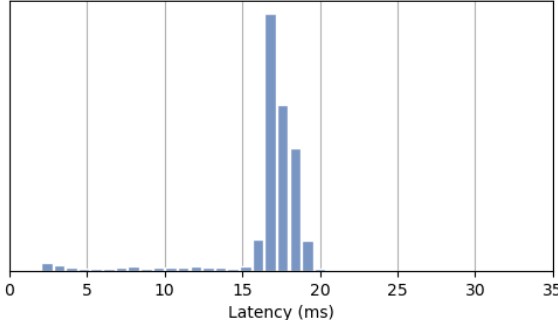

Figure A: Runtime distribution on EuRoC. Our system averages 60FPS with each new frame taking ~17ms to process. The distribution is very centralized and rarely drops below 50FPS.

## B  ATE Error Metric

We compare results using the ATE (average trajectory error) metric, which is standard for VO and SLAM (16; 37; 27; 43; 12). The ATE metric between the predicted trajectory $\tilde{\mathbf{T}}$ and the ground-truth trajectory $\hat{\mathbf{T}}$ is computed using

$$E(\tilde{\mathbf{T}}, \hat{\mathbf{T}}) := \sum_{t=1}^{N} ||\tilde{\mathbf{T}}_{xyz} - \hat{\mathbf{T}}_{xyz}||_2 \tag{9}$$

after aligning the two trajectories using a similarity transformation, to account for the scale and SE(3) gauge freedoms. This metric is computed using the EVO library (18).

## C  Additional Training Details

DPVO is trained entirely on TartanAir (44)[2], a synthetic dataset. This is the same synthetic training data used by previous VO systems (43; 37). TartanAir provides a large number of training scenes of both indoor and outdoor environments with varied lighting and weather conditions. The dataset provides depth and camera pose annotations, enabling one to generate optical flow by re-projecting

---

[2]https://theairlab.org/tartanair-dataset/

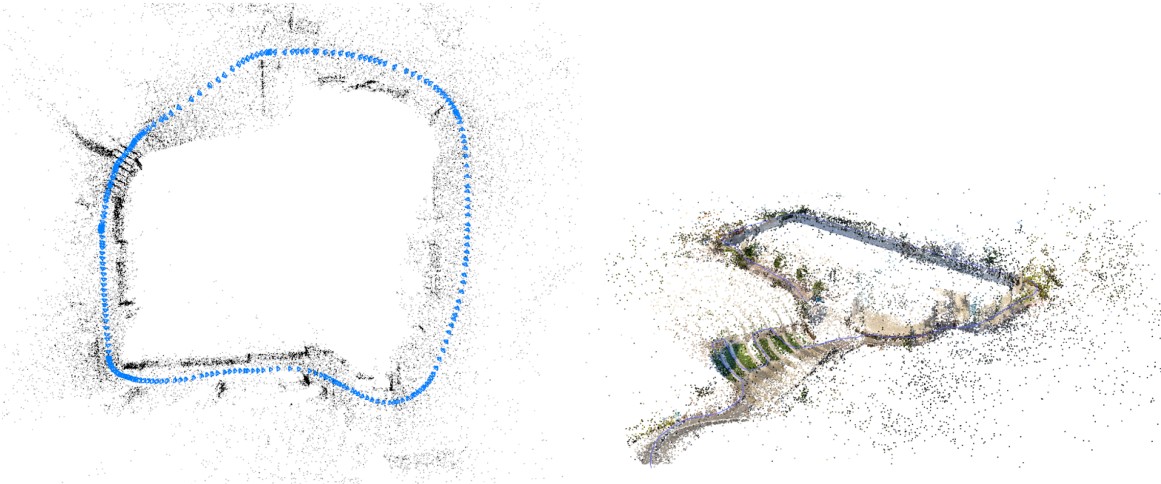

(a) Reconstruction on the ETH-3D (32) dataset.    (b) Reconstruction from our video demo.

Figure B: Sparse reconstructions produced by our model.

the depth using the camera poses. All dataset parameters are identical to those used in DROID-SLAM (37). This includes random cropping, color-jitter, random greyscale, random color inversion, as well as re-scaling the training scenes in order to improve training stability.

# D  Visualization

Reconstructions are visualized interactively using a separate visualization thread. Our visualizer is implemented using the Pangolin library[3]. It directly reads from PyTorch tensors avoiding all unnecessary memory copies from CPU to GPU. This means that the visualizer has very little overhead–only slowing the full system down by approximately 10%. In Fig. Ba, we show a reconstruction on the ETH-3D (32) dataset, and in Fig. Bb we show the reconstruction produced during our video demo.

# E  Bundle Adjustment Layer

Our bundle adjustment layer is functionally identical to the BA layer in DROID-SLAM (37), except that DROID-SLAM predicts the damping factor using its update operator, while we found this not to be necessary and hard-code it to $10^{-4}$ instead. We refer the reader to Appendix C in the DROID-SLAM paper for the analytical gradients of the Gauss-Newton update iterations. Since DROID-SLAM uses dense flow, we implement our own sparse version as an optimized CUDA layer.

# F  Additional Figures

**Correlation Operation:** In Fig. Ca, we show a high-level visualization of the correlation operation which is performed at the start of the update operator. Patch $k$'s $P^2$ matching-features are each projected into the two-level feature pyramid of frame $j$. A $7 \times 7$ grid of $D = 128$ dimension feature vectors are bilinearly sampled from the pyramid around each point reprojection and dot producted with the matching features. This operation results in 2 sets of $p \times p \times 7 \times 7$ dot products, one for each resolution (1/4 and 1/8th of the original image size). This operation is implemented as an optimized CUDA layer.

**Update Operator:** Following Fig. Ca, in Fig. Cb we show a high-level view of the rest of the update operator. The correlation and context features are fed into additional learn-able layers, including 1D

---

[3]`https://github.com/stevenlovegrove/Pangolin`

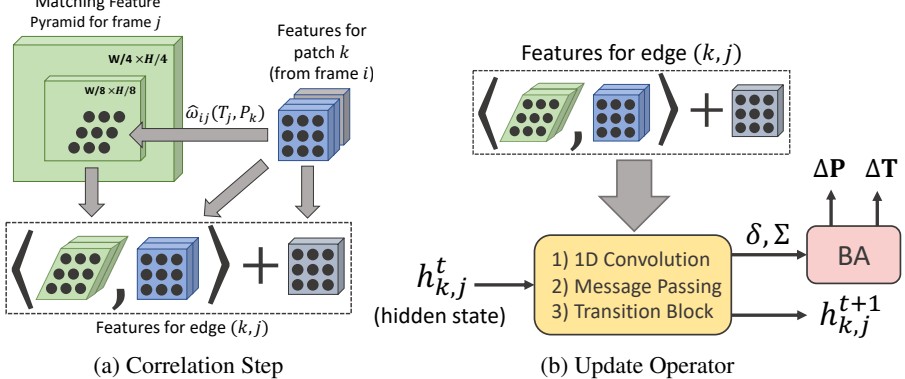

(a) Correlation Step            (b) Update Operator

Figure C: The update operator, including the correlation step. **(a)** For each edge $(k, j)$ in the patch graph, patch $k$ is reprojected into a frame $j$ using equation (2). Frame $j$'s matching features are then cropped using the reprojected patch. The *correlated* matching features (blue, green) and context features (grey) form the edge features. Each dot in the sampled matching features (green) represents a $7 \times 7$ neighborhood of points, however this is omitted for clarity. **(b)** Given the features for edge $(k, j)$, several learnable layers (yellow) are used to predict an update $\delta_{(k,j)} \in \mathbb{R}^2$ to the flow, a confidence weight $\Sigma_{(k,j)} \in \mathbb{R}^2$, and an update to the hidden state $h_{k,j}^t$. The predicted factors are used in the bundle adjustment layer to produce an update to the camera poses and patch depths. The factor head is omitted here for clarity.

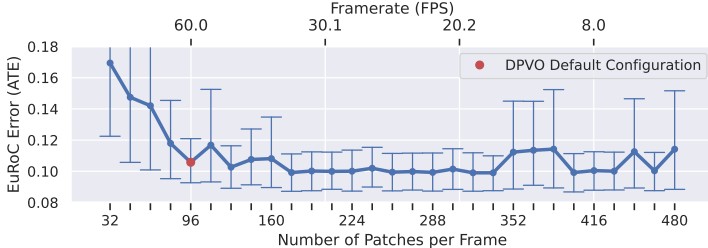

Figure D: Accuracy on the EuRoC dataset as a function of the number of patches. We can trade-off speed for performance by increasing the number of patches tracked. However, performance quickly saturates beyond 96.

Convolution Layers, a Softmax-Aggregation block (i.e. Message Passing), and a Transition Block. This results in an updated hidden state for each edge. Finally, the updated hidden state is used to produce 2D flow revisions and confidence weights which guide the solution to the bundle adjustment. The bundle-adjustment layer ultimately produces an update to the depths and camera poses.

**Confidence Weights:** Our approach learns to reject outliers by predicting a confidence weight associated to each predicted 2D flow update, which DROID-SLAM (37) does as well. We show a visualization of the weights for a small number of edges in Fig. E. These weights are not supervised directly, rather they are learned by supervising on the pose output of the differentiable bundle-adjustment layer.

**Number of Patches:** In Fig. D, we show the accuracy and runtime of DPVO on EuRoC while varying the number of patches. Results are accumulated across 5 independent trials. The accuracy quickly saturates after 96 patches.

## G    Network Architecture

**Update Operator:** In Fig. F, we show the architecture of the update operator, excluding non-learnable layers such as the correlation layer and the bundle adjustment layer. Since the 1D-Convolutions and the Softmax-Aggregation layers operate on neighoring edges, they are partially implemented using PyTorch (28) scattering operations.

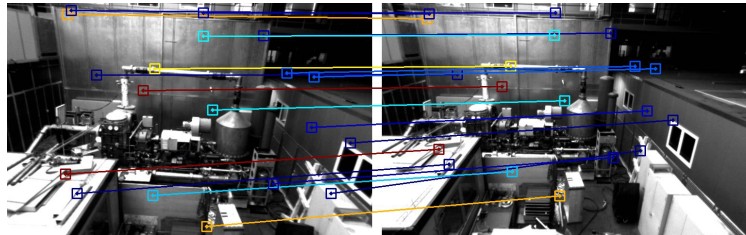

Figure E: Factor confidence weights. For each edge in the patch graph, the factor-head of our update operator predicts a confidence weight $w \in \mathbb{R}^2$ bounded to $(0, 1)$ and a 2D flow update. Cold colors (blue) represent high confidence edges while hot colors (red) represent low confidence edges.

**Feature Extractor:** In Fig. G, we show a visualization of the architecture of the feature extractor. The same network architecture is used for extracting context features and for extracting matching features, except the former uses no normalization and output dimension $D = 384$ while the latter uses instance normalization and $D = 128$. We use instance normalization for the matching features since they should be calculated independently for each input image in a batch, which other flow-based networks (36; 38; 37) do as well. Our feature extractor is similar to the architecture used in DROID-SLAM (37), but half the size. Consequently, the output resolution is 1/4 of the image resolution, as opposed to 1/8th. Since DPVO only tracks a sparse collection of patches instead of predicting dense flow, we can afford to use higher spatial-resolution features without significant memory overhead.

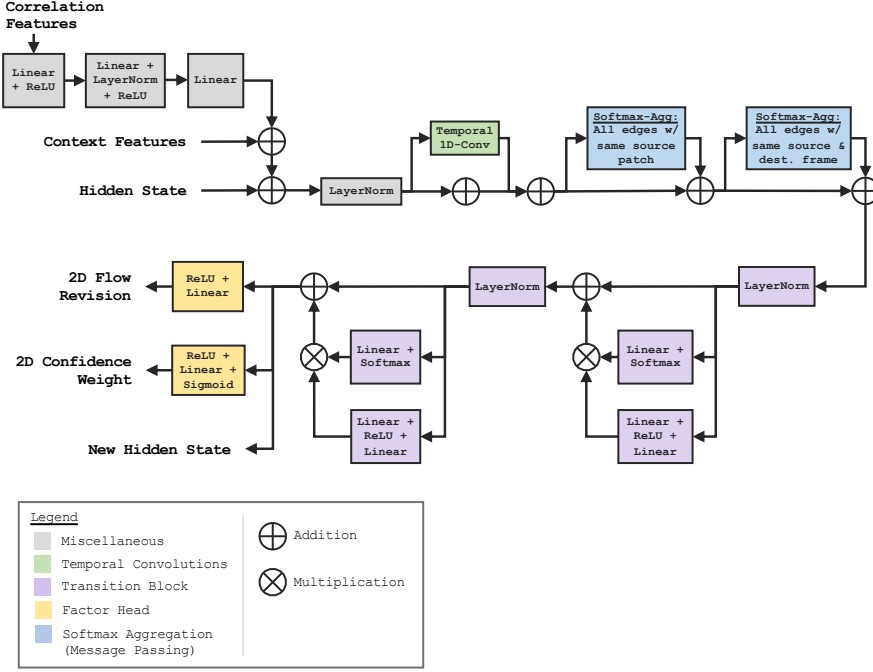

Figure F: Architecture of the Update Operator, excluding non-learnable layers such as the correlation layer and the bundle-adjustment layer.

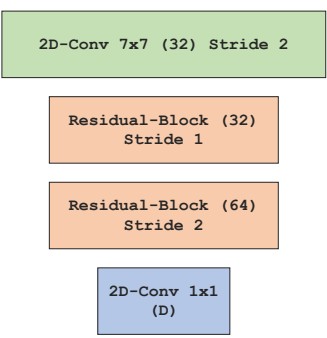

Figure G: Architecture of the feature extractors. $D = 128$ for the matching-feature extractor and $D = 384$ for the context-feature extractor.

# H Predicted Trajectories on EuRoC

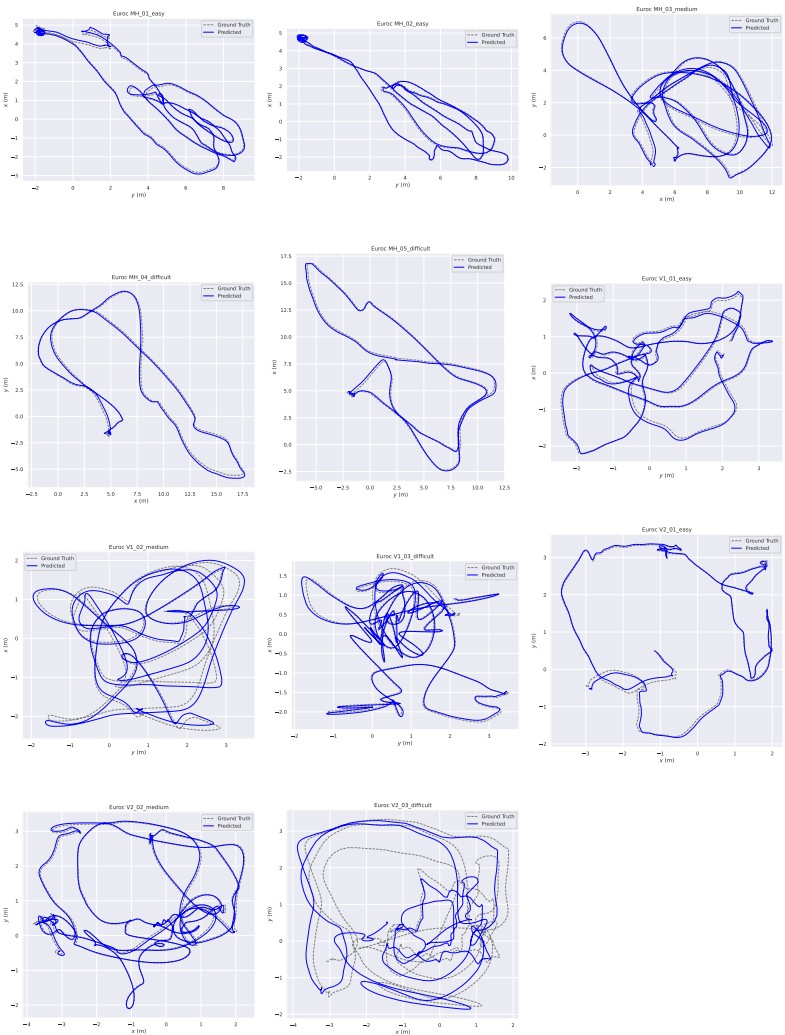

Figure H: Trajectory predictions on the EuRoC test dataset.

# I  Predicted Trajectories on TartanAir

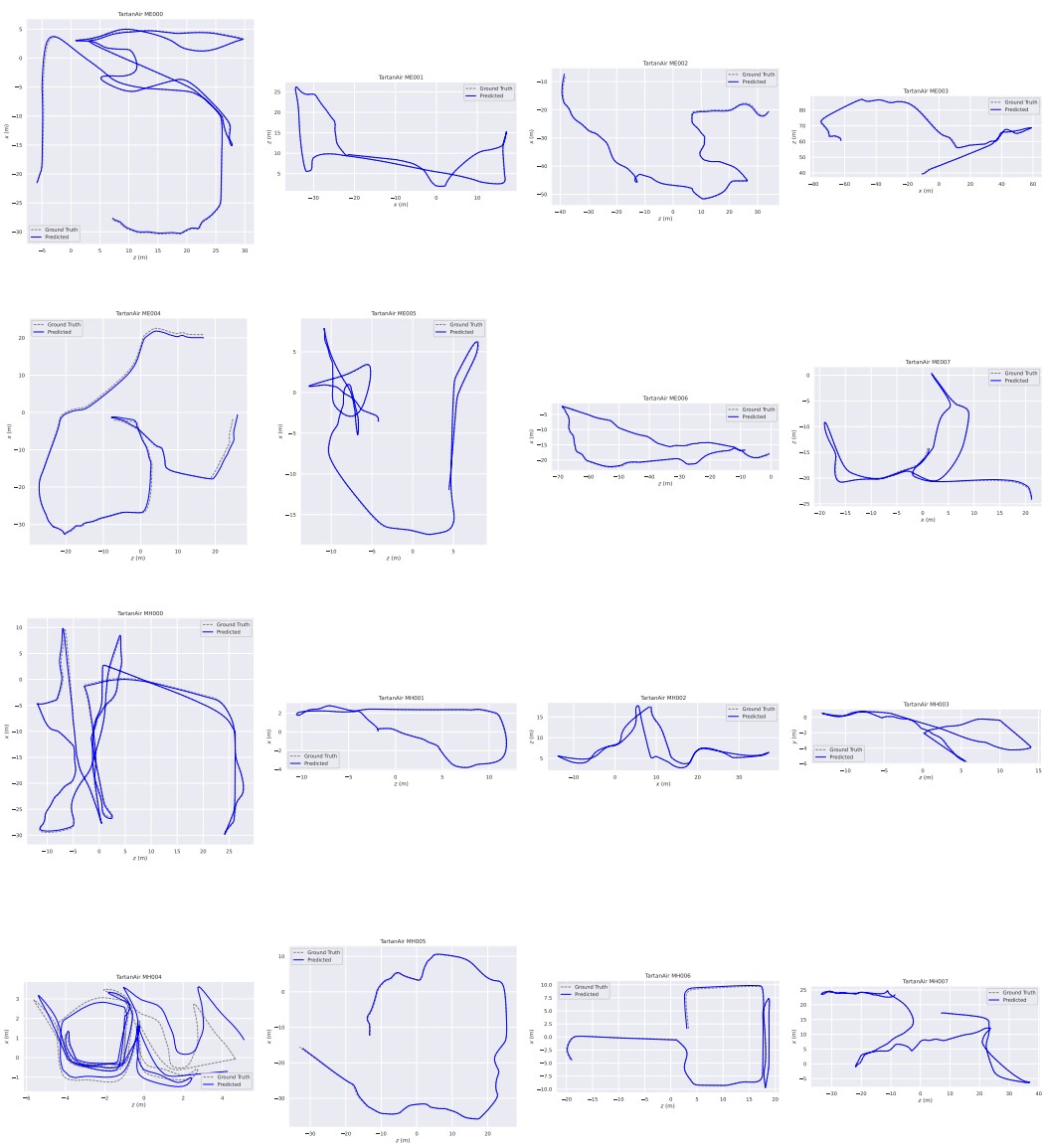

Figure I: Trajectory predictions on the TartanAir test dataset.

