# OpenReview forum: "Deep Patch Visual Odometry"
_NeurIPS.cc/2023/Conference — NeurIPS 2023 poster_

### Official Review · Reviewer_kzhF · 2023-06-27

**Soundness:** 3 good
**Presentation:** 2 fair
**Contribution:** 2 fair
**Rating:** 5
**Confidence:** 3

**Summary:**

This paper proposes a deep learning-based method for monocular visual odometry, which equips two leading-edge advantages: (i) a deep feature-based patch representation for keypoints encoding local context, and (ii) a novel recurrent architecture designed for patches along with a differentiable bundle adjustment layer.  The proposed DPVO outperforms the state-of-the-art across several common benchmarks while running 1.5-8.9x faster than the DROIOD-SLAM system with less memory leveraged.

**Strengths:**

The proposed monocular visual odometry performs accurately in estimating the camera pose in several datasets. And the proposed system shows a strong generalization that is trained on the synthetic TarTanVO dataset but validated on real datasets.

**Weaknesses:**

1. The paper makes claims that the proposed method outperforms all prior work in several evaluation datasets, but only several works are listed which is not enough, can authors give more comparisons with SOTA? For example, the DytanVO[1], RAM-VO[2], DF-VO[3]

[1]https://arxiv.org/pdf/2209.08430v4.pdf

[2] https://arxiv.org/pdf/2107.02974v1.pdf

[3] https://arxiv.org/pdf/2103.00933v1.pdf

2. The results on various datasets are measured by ATE(Absolute Trajectory Error), but some other evaluation criteria are also essential, for example, the relative pose error (RPE), the average translational error, and the rotational error. Can the author provide more comparison information about them?

3. The tracking of patches is achieved by designing an RNN but fails to provide a clear explanation. Why not a CNN?

4. Punctuations are necessary at the end of each equation. Please carefully check it.

**Questions:**

See Weakness part

**Limitations:**

The paper claims that the current learning-based VO systems are impractical for resource-constrained devices. However, the proposed method still cannot be applied to mobile GPU-free devices.

---

> ### Author Rebuttal · Authors · 2023-08-10
>
> **The paper makes claims that the proposed method outperforms all prior work in several evaluation datasets, but only several works are listed which is not enough, can authors give more comparisons with SOTA? For example, the DytanVO[1], RAM-VO[2], DF-VO[3]**
>
> Our claim was intended to mean that we outperform all prior work that is known to us and has reported results on the 4 datasets, which are standard benchmarks for VO/SLAM. We will clarify this point in our revision.
>
> We include an additional Table F in the rebuttal PDF of results comparing DPVO against DF-VO [Zhan et al. ‘20]. We observe that DPVO outperforms DF-VO on average.
>
> Unfortunately, we do not yet have direct comparisons with DytanVO and RAM-VO. DytanVO and RAM-VO reported no results on any of the 4 datasets we use.  To use their open-source code requires us to develop nontrivial dataloading code for each dataset. We are in the process of doing so, but have not been able to complete it before the deadline.
>
>
> **The results on various datasets are measured by ATE(Absolute Trajectory Error), but some other evaluation criteria are also essential, for example, the relative pose error (RPE), the average translational error, and the rotational error. Can the author provide more comparison information about them?**
>
> Table F in the rebuttal PDF includes comparisons using relative pose error (RPE). It shows that our method is superior also in terms of RPE.
>
> We primarily used the ATE metric for evaluation in the main paper since this enabled us to easily compare DROID-VO/SLAM and ORB-SLAM on more datasets, such as TartanAir, TUM-RGBD, and EuRoC.
>
> We agree that more metrics is always better, however we evaluated using ATE as it is the primary metric used in several prior works [DROID-SLAM, ORB-SLAM3], which allowed us to compare to prior works on a large number of datasets without the need to reproduce the results ourselves. It is worth pointing out that ATE is popular because the final camera positions (which ATE evaluates) are effectively a function of the accumulated error in both translation magnitude and direction (which is related to the predicted rotation), i.e. it’s extremely unlikely to have very low ATE but high rotation error.
>
> **The tracking of patches is achieved by designing an RNN but fails to provide a clear explanation. Why not a CNN?**
>
> The update operator is recurrent (i.e. weight-tied) since it continually improves its own predictions based on its current state through repeated applications. A CNN would not be a natural choice. Some explanations are already provided on L147-L158 + Figure 2 in the main paper and L53-L58 + Figure C in the supplement, but we will revise and clarify.  As mentioned in L171-L177, one component of DPVO’s update operator is a 1D convolution across the temporal dimension of each patch trajectory. Therefore, our update operator is an RNN with a CNN layer inside it.

---

> > ### Comment · Reviewer_kzhF · 2023-08-11
> >
> > Thanks for your rebuttal. I am satisfied with your reply.

---

### Official Review · Reviewer_5ejm · 2023-06-29

**Soundness:** 3 good
**Presentation:** 2 fair
**Contribution:** 2 fair
**Rating:** 6
**Confidence:** 3

**Summary:**

The author propose a learning-based approach for visual odometry. It is essentially a sparse version of DROID-SLAM, without the loop closure capability. Instead of predicting the dense flow field relating the different frames as in DROID-SLAM, DVPO operates on sparse image patches, which are extracted at random from each frame. In more details, a random set of patches are extracted from each video frame, a patch graph is built. A recurrent neural network predicts the residual motion of each patch in each other frame. These residual motions are then used in a differentiable BA layer which minimizes the reprojection error to optimize for the sparse depth of each patches and the camera poses. This process is repeated iteratively until convergence. The paper proposes a new architecture to predict the residual motion, adapted to patches.

**Strengths:**

1) Thorough and convincing evaluation: The proposed approach leads to a convincing improvement in terms of speed and accuracy, compared to DROID-SLAM. The approach is evaluated on many different datasets.

2) The ablation study also pin-points the essential contributions

**Weaknesses:**

In DROID-SLAM, predicting the dense flow allowed to optimize for full dense depths along with the camera poses with accuracy, providing advantages over both direct and indirect approaches. However, the proposed approach DVPO is based on jointly estimating the motion of sparse patches and refining camera poses and sparse depth (of the patches).
In that sense, it is not obvious what advantages are brought by coupling the matches estimation with the pose/depth prediction. Classical indirect approaches first predict sparse matches, triangulate 3d point to get the depth (thus providing initialization for the depth of ‘patches’), and then track further frame by performing PnP with multiple local and global BA. The paper compares to ORD-SLAM and outperforms it. However, there are now many sparse matching approaches that significantly outperform hand crafter detectors like ORB. How would DVPO compare to using a ‘classical’ indirect approach with matching performed by a deep-learning based state-of-the-art approach like SuperPoint-SuperGlue?


**Questions:**

1) see weakness

2) A better figure for the patch graph would be help the understanding, showing what are the nodes, what are the edges, where the updates are and so on. Fig.1 is not very informative. It would significantly help to read sec. 3.1.
I also find using the same index notation for the patches and for the images very confusing. ex L.159 (k, i) refers to an edge between patch k and image I. It would make it easier if the patches had a different notations like bold k.

3) some implementation details: how large are the patches?

4) in the ablation study, it is shown that using a patch is better than point feature. According to the formulation for estimating the motion of the patch, using a patch effectively increases the search window of the correlation in the other image. Does this difference still hold if the point feature is correlated to a larger window in the other image to estimate its motion?


**Limitations:**

yes

---

> ### Author Rebuttal · Authors · 2023-08-10
>
> **In DROID-SLAM, predicting the dense flow allowed to optimize for full dense depths along with the camera poses with accuracy, providing advantages over both direct and indirect approaches. However, the proposed approach DVPO is based on jointly estimating the motion of sparse patches and refining camera poses and sparse depth (of the patches). In that sense, it is not obvious what advantages are brought by coupling the matches estimation with the pose/depth prediction.**
>
> The advantages of DPVO over other methods come from a combination of (1) sparse matches, (2) optimizing pose and depth (bundle adjustment) on sparse matches, (3) coupled iterations of sparse matches and bundle adjustment, (4) end-to-end differentiability and training.
> It is true that (1)  + (2) are also used by many existing methods, particularly indirect methods. Thus it may appear that DPVO has lost its advantages by using sparse matches. However, the full combination of (1) to (4) is unique to DPVO. In other words, the advantages of DPVO over indirect methods come from integrating (3) + (4) with (1) + (2).
>
> **Classical indirect approaches first predict sparse matches, triangulate 3d point to get the depth (thus providing initialization for the depth of ‘patches’), and then track further frame by performing PnP with multiple local and global BA. The paper compares to ORD-SLAM and outperforms it. However, there are now many sparse matching approaches that significantly outperform hand crafter detectors like ORB. How would DVPO compare to using a ‘classical’ indirect approach with matching performed by a deep-learning based state-of-the-art approach like SuperPoint-SuperGlue?**
>
> In table E in the rebuttal PDF, we show that DPVO achieves 62% lower error on TartanAir than an approach based on COLMAP with SuperGlue+Superpoint matching, while running 80x faster. This superglue-based result is also reported in the DROID-SLAM paper. We will add this comparison to our next revision.
>
> The comparison is on the TartanAir test set, following the evaluation in the ECCV 2020 SLAM competition and in [DROID-SLAM]. The score is computed using normalized relative pose error for all possible sequences of length {5, 10, 15, ..., 40} meters.
>
> The Superglue + Superpoint approach runs 40x slower than real-time, as is expected since this approach is designed for offline structure-from-motion, not visual odometry.
>
> **A better figure for the patch graph would be help the understanding, showing what are the nodes, what are the edges, where the updates are and so on. Fig.1 is not very informative. It would significantly help to read sec. 3.1. I also find using the same index notation for the patches and for the images very confusing. ex L.159 (k, i) refers to an edge between patch k and image I. It would make it easier if the patches had a different notations like bold k.**
>
> In Figure L in the rebuttal PDF, we’ve included a new figure illustrating a single edge in the patch-graph connecting two nodes (a frame and a patch), and a hypothetical update that could be made to that edge. Thank you for the suggestions, we will include them in our revision.
>
> **Some implementation details: how large are the patches?**
>
> They are 3 x 3 at ¼ the image resolution, so they effectively cover 12 x 12 pixels in the original image.
>
> **In the ablation study, it is shown that using a patch is better than point feature. According to the formulation for estimating the motion of the patch, using a patch effectively increases the search window of the correlation in the other image. Does this difference still hold if the point feature is correlated to a larger window in the other image to estimate its motion?**
>
> This is an empirical question on the combination of two hyperparameters: patch size and correlation radius. While we found the patch size 3x3 to work better than 1x1 while keeping other hyperparameters constant (as is common practice for hyperparameter ablations), we did not exhaustively test all combinations so as to avoid overfitting to the validation set.
>
> Due to an initial misunderstanding of this question (we mistook it to be a conceptual question rather than empirical), we did not start the necessary experiments early enough to be able to report this specific ablation by the rebuttal deadline, but we will post the result as soon as we are able.

---

> > ### Comment · Reviewer_5ejm · 2023-08-18
> >
> > Thank you for your answer. I agree with other reviewers about the limited novelty of the paper. Nevertheless, I also agree with the authors that providing a fast and as accurate alternative is interesting. For this reason, i will maintain my score. But i also won't oppose if the paper is not accepted.

---

> ### Author Response · Authors · 2023-08-21
> **Response to Question #4**
>
> Thank you for your comment.
>
> As promised, we provide the requested empirical study in response to question #4. Specifically, we re-trained a second DPVO model (hence the delay) using point features instead of patches, but with the features correlated to a larger window (radius 8, instead of 7) in the other image to match the effective search window of 3x3 patches, and compared the models on the “hard” sequences of the TartanAir test set.
>
> The conclusion in the following table is the same as in our ablations: patch features still significantly outperform point features overall, even when the latter is correlated to a larger window in the other image to compensate. We will include this comparison in our revision.
>
> | _Method_ | MH000      | MH001 | MH002 | MH003  | MH004 | MH005 | MH006      | MH007 | Avg
> | ----------- | ----------- | ----------- | ----------- | ----------- | ----------- | ----------- | ----------- | ----------- | ----------- |
> | DPVO (Patches, Grid-radius=7) | **0.21** | **0.04** | **0.04** | **0.08** | **0.58** | **0.17** | **0.11** | **0.15**  | **0.173** |
> | DPVO (Points, Grid-radius=8)  | 0.73 | 0.05 | 0.09 | 0.09 | 0.76 | 0.52 | 0.14 | 0.24  | 0.328

---

### Official Review · Reviewer_5bXL · 2023-06-30

**Soundness:** 2 fair
**Presentation:** 3 good
**Contribution:** 2 fair
**Rating:** 3
**Confidence:** 5

**Summary:**

In this paper, the author propose a new method to solve monocular visual odometry (VO) in and end-to-end fashion. In contrast to previous work - DROID-SLAM, which utilizes dense optical to build correspondences across frames, the proposed method leverages sparse patches to avoid redundancy of dense pixels and therefore works faster. Experiments on public datasets including TartanAir, TUM-RGBD, EuRoC and ICL-NUIM demonstrate its higher efficiency and competitive accuracy.

**Strengths:**

1.	Good motivation. Dense correspondences provided by optical flow estimation are not necessary for visual odometry because basically several good matches are enough to find a good pose (more matches may increase the robustness). Therefore, I agree with the author of using sparse patches to achieve higher time and memory efficiency.
2.	Extensive results. The proposed method is evaluated on TartanAir, TUM-RGBD, EuRoC and ICL-NUIM datasets to show its efficiency and accuracy.
3.	The paper is well-written and easy to read.

**Weaknesses:**

1.	Limited novelty. The key idea of the paper is relatively straightforward. If we review the classic monocular visual VO/SLAM systems, we can find most of them are based on sparse keypoints (ORB-SLAM) or semi-dense pixels (DSO) to avoid too much computation. From my point of view, the major contribution of the paper comes from the engineering part. The recurrent module designed to update the trajectories of patches is kind of novel as it automatically updates the covisibility graph constructed by observed patches and past frames. However, the overall novelty is still limited.
2.	Related works. Another limitation is the discussion of previous works. In addition to DROID-SLAM, there are lots of excellent learning-based VO/SLAM systems proposed in the past a few years such as BeyondTracking [r1], NICE-SLAM [r2], iMAP [r3], and Li et al [r4] to name a few. These works have achieved SOTA performance. However, they are not discussed and compared neither. It would be better to give a discussion of these works in the paper.

r1: Xue et al., Beyond Tracking: Selecting Memory and Refining Poses for Deep Visual Odometry.CVPR 2019

r2: Zhu et al., NICE-SLAM: Neural Implicit Scalable Encoding for SLAM. CVPR 2022

r3: Sucar et al., iMAP: Implicit mapping and positioning in real-time. ICCV 2021

r4: Li et al., DENSE RGB SLAM WITH NEURAL IMPLICIT MAPS. ICLR 2023.


**Questions:**

1.	L141: random sampling. It is not very clear that random sampling of patches works better. Intuitively, with the predicted camera motion, sampling according to the predicted pose could give more correct matches between patches. Even in the classic VO/SLAM systems like ORB-SLAM and DSO, they sample keypoints/pixels from regions with rich textures because these regions could be better observed in following frames. The comparisons of random sampling with other strategies could solidate the claim.

2.	In the Differential Bundle Adjustment module, the cameras poses and inverse depths are optimized with pixel coordinates fixed. In practice, the matches between patches across frames have outliers, impairing the performance. Basically, the BA and outlier removing are jointly updated in classic systems to guarantee the accuracy. It seems that this can be achieved in the recurrent module and as this step is very important especially to monocular VO/SLAM systems, it would be great to show how the module ‘corrects’ wrong matches.

---

> ### Author Rebuttal · Authors · 2023-08-10
>
> **Limited novelty. The key idea of the paper is relatively straightforward. If we review the classic monocular visual VO/SLAM systems, we can find most of them are based on sparse keypoints (ORB-SLAM) or semi-dense pixels (DSO) to avoid too much computation. The recurrent module designed to update the trajectories of patches is kind of novel as it automatically updates the covisibility graph constructed by observed patches and past frames. However, the overall novelty is still limited.**
>
> Regarding the novelty of our approach, please see our message to everyone in which we highlight why using sparse matches with the framework of DROID-SLAM is novel and challenging.
>
> **From my point of view, the major contribution of the paper comes from the engineering part.**
>
> We believe a well-engineered system is still a valuable contribution to NeurIPS, especially given our exceptionally fast runtime, low cost, and performance on-par or better than DROID.
>
>
> **Related works. Another limitation is the discussion of previous works. In addition to DROID-SLAM, there are lots of excellent learning-based VO/SLAM systems proposed in the past a few years such as BeyondTracking [r1], NICE-SLAM [r2], iMAP [r3], and Li et al [r4] to name a few. These works have achieved SOTA performance. However, they are not discussed and compared neither. It would be better to give a discussion of these works in the paper.**
>
> **r1: Xue et al., Beyond Tracking: Selecting Memory and Refining Poses for Deep Visual Odometry. CVPR 2019**
>
> **r2: Zhu et al., NICE-SLAM: Neural Implicit Scalable Encoding for SLAM. CVPR 2022**
>
> **r3: Sucar et al., iMAP: Implicit mapping and positioning in real-time. ICCV 2021**
>
> **r4: Li et al., DENSE RGB SLAM WITH NEURAL IMPLICIT MAPS. ICLR 2023.**
>
>
> We did not compare to r1, r2, r3, or r4 for several reasons:
> 1) r2 and r3 assume RGB-D input, whereas DPVO is RGB-only.
> 2) r1 trains on TUM-RGBD sequences that share the same scenes with the test sequences, whereas DPVO is evaluated on TUM-RGBD zero-shot without any fine-tuning.
> 3) r2, r3 and r4 use global optimization and are not typically considered VO methods.  Following standard practice [e.g., TartanVO ‘21], we focused our comparisons on strictly VO methods, i.e., those that do not perform any global optimization: Global optimization can substantially improve results on input sequences with loops, but at the expense of speed.
>
> Regardless, we’ve included comparisons of DPVO against r1, r2, r3 and r4 in tables B,C and D in the rebuttal PDF.
>
> *Comparison to r2, r3, r4 on TUM-RGBD*:
>
> We compare to r2, r3 and r4 in Table B on the popular [fr1/desk fr2/xyz fr3/office] split of the TUM-RGBD dataset.  These sequences have been used for comparisons by multiple papers including iMAP and NICE-SLAM. These results show that our approach can outperform others on some sequences, even when they use additional depth sensor input, or use global optimization.
>
> In Table B, we observe that DPVO outperforms all methods on fr2/xyz, and underperforms Li et al. on fr1/desk. DPVO underperforms others on fr3/office, which is expected because the entire fr3/desk sequence is a loop, which favors methods with global optimization. DPVO with default settings runs nearly 3x faster than all the three, and our model runs more than 8x faster in its “fast” configuration.
>
> r3 (iMAP) and r2 (NICE-SLAM) assume an RGB-D input sequence, while r4 (Li et al.) and our approach do not require depth, only RGB input.
>
> Li et al. and NICE-SLAM do not globally optimize all poses, but still use the global keyframe list and/or the global 3D map to optimize the keyframe poses in the optimization window.
>
> DPVO also has near-constant memory usage in all cases, as opposed to iMAP, NICE-SLAM, and Li et al. whose memory grows proportionally to the sequence length in an unbounded scene. Unlike these other approaches, DPVO’s memory is always bounded; The RGB sequence could go on forever and DPVO would never run out of memory because it never touches old keyframes or the old 3D map.
>
>
>
>
> *Comparison to r4 on EuROC*:
>
>
>
>
>
>
> We again compare our VO system to r4 (Li et al.) on the EuRoC test dataset in Table C in the rebuttal PDF. Li et al. outperforms DPVO on 2 / 6 of the sequences, catastrophically fails on another 2 / 6 of the remaining sequences, and underperforms DPVO on the remaining 2 / 6.
>
> *Comparison to r1 on TUM-RGBD (separate split)*:
>
> We compare to r1 (Xue et al.) in table D, on a separate split of TUM-RGBD used for evaluation by Xue et al, since they did not report results on the sequences used by r2-r4. DPVO outperforms Xue et al. on three of the sequences, while they outperform us on the remaining sequences.
>
> It is important to note that Xue et al. is trained on sequences that share the same scenes with the test sequences, whereas our approach is trained only on synthetic data and is tested on TUM-RGBD zero-shot.
>
>
> **Response to Q1**
>
> These comparisons were already provided in figure 5c in the main paper, where we show evidence that random sampling does indeed work better than many other approaches. One of these approaches is sampling areas with high image gradients, which is analogous to sampling from pixels with rich textures.
>
> Note that selecting patch centroids randomly is the simplest possible approach (no keypoint detector required). We do not claim that it is the optimal choice, but the fact that such a simple scheme works well is a strength, not a weakness.
>
> **Response to Q2**
>
> Our method does indeed reject outliers in the recurrent module, by predicting a confidence weight associated with each predicted match. These confidences are then treated as constants in the BA step. This enables DPVO to exclude unlikely matches from the optimization entirely, without needing to correct them.
>
> We follow DROID-SLAM in this regard, and don’t claim this as a contribution of our method. We show an example of the predicted per-match confidence in Fig E of the supplement.

---

> > ### Comment · Reviewer_5bXL · 2023-08-14
> > **rebuttal**
> >
> > Thanks for the responses. The newly added comparisons with previous VO methods (r1, r2, r3, r4) make the results of the proposed method more convincing. I don’t have any more questions about the experiments and hope the analysis and discussions of these results can be included in the refined manuscript.
> >
> >
> > My major concern, however, is still the novelty. As claimed by the author in the rebuttal and also mentioned by the reviewer Uwmu, the major contribution is to use randomly sampled patches to replace dense optical flow in DROID-SLAM to achieve higher efficiency. I don’t see more novelty from this design in spite of the better performance which could be achieved from the engineering.

---

> > > ### Author Response · Authors · 2023-08-14
> > > **Response to reviewer 5bXL**
> > >
> > > Thank you for your comment. Regarding novelty, although sparse matches in itself was not new, it was not obvious whether it would work at all in the framework of DROID-SLAM, because the SOTA performance of DROID-SLAM was understood to be dependent on the dense flow. While it may look simple and obvious in hindsight, making sparse matches work in DROID-SLAM was far from straightforward, for reasons we detailed in the global message.
> > >
> > > In addition to novelty, the value of this work also lies in a practical, open-source system that achieves the same level of accuracy and runtime that was only achievable with much more computing resources. DROID-SLAM required a large-memory GPU to be real time, making it impractical for many applications. DPVO achieved the same or better accuracy while being much faster and memory efficient.
> > >
> > > Building well-engineered practical systems is an important contribution in the space of VO/SLAM. We believe this type of contribution should not be trivialized because online, real-time, resource-efficient processing is a basic requirement that differentiates the VO/SLAM task from other 3D reconstruction tasks.
> > >
> > > We believe that even though the perception of novelty can vary, a practical open-source system that achieves important, previously unavailable capabilities can be sufficiently valuable to the NeurIPS community.

---

### Official Review · Reviewer_Uwmu · 2023-07-06

**Soundness:** 3 good
**Presentation:** 4 excellent
**Contribution:** 2 fair
**Rating:** 5
**Confidence:** 4

**Summary:**

This paper proposes a deep learning based visual odometry (VO) system which estimates the 6DoF poses of each frame and outputs a sparse reconstruction of the scene from an input monocular video sequence. The pipeline extracts deep features from the sampled image patches and updates deep optical flow of the patches with a recurrent module, then perform deep BA with patch correspondences within the local optimization windows on the fly. The reported statistics show that this system is able to run at 60FPS (Default) and 120FPS (Fast), respectively, while maintaining the camera pose estimation accuracy. Results show that the proposed method outperforms DROID-VO and ORBSLAM in terms of accuracy on TARTAN AIR, EuRoC MAV, and ICL-NUIM datasets.

**Strengths:**

1. One of the main contribution of this work is to alleviate the large computational cost from the existing deep dense VO method (DROID-SLAM) while maintaining its pose estimation accuracy.

2. The paper shows the contribution of 1D Temporal Convolution and SoftMax Aggregation on neighboring frames to the pose estimation, which are novel.

3. The learned network is trained purely on synthetic data and it shows competency to perform well on the real datasets, including TUM-RGBD and EuRoC MAV.

4. The paper is clearly written and easy to follow.

**Weaknesses:**

1. The novelty is limited, the iterative update mechanism and differential bundle adjustments are already used in DROID-SLAM, even with the introduction of 1D Temporal Convolution and SoftMax Aggregation modules in the update operator.

2. Another major contribution of the proposed method is to track image patches instead of using dense correspondence on the entire image. However, it is not theoretically sound to support the claim that the patch-based approach improves the accuracy of pose estimation results over the dense flow based approach when the patches are selected randomly rather than feature based methods.

3. I suspect that the model overfits the TartanAir dataset. That’s why there is a big performance gap between the results on synthetic and real datasets for the classic methods. When it comes to real datasets, the results are mixed.

4. The selection of baseline methods is inconsistent and questionable in Table 1-4, the ORB-SLAM version used is different, and authors did not compare the results to ORB-SLAM or D3VO on the EuRoC MAV dataset, while other VO papers do include them as baseline methods. Also, there are better results can be found for running ORB-SLAM on TUM-RGBD fr1 and TartanAir sequences (reported in https://paperswithcode.com/paper/droidslam-deep-visual-slam-for-monocular/review/).

**Questions:**

1. Are there any insights for the reasons of randomly sampled patches outperforming other feature based patch selection mechanisms (ORB, SIFT, and Superpoint)? Although the empirical results in this paper favor randomly sampled patches, it is still counter-intuitive to imagine most patches are randomly sampled on regions with repetitive patterns or occlusion. Are there any additional experiments to demonstrate the robustness of random patch selection?

2. As mentioned in Weaknesses, why ORB-SLAM or D3VO are not included when comparing the results on the EuRoC MAV dataset? Are these methods not considered visual odometry systems?

3. Both DROID-SLAM and DVPO are trained from synthetic datasets, are there any suggestions on the reasons for the worse performance of DROIDSLAM than DPVO on the ICL-NUIM dataset, even with global bundle adjustment?

4. Are the running parameters (window size, patch size, number of update iterations) of the system tuned for different datasets, or are they fixed? Does patch size matter (not including the 1x1 point feature)?

5. Are there any comparison between the system and classical methods in terms of efficiency, with different input frame sizes?

**Limitations:**

The model can only be trained on synthetic datasets with ground truth optical flow. Although the system shows the ability of generalization over the real data, its performance and robustness may be limited in the real scenarios.

---

> ### Author Rebuttal · Authors · 2023-08-10
>
> **The novelty is limited, the iterative update mechanism and differential bundle adjustments are already used in DROID-SLAM, even with the introduction of 1D Temporal Convolution and SoftMax Aggregation modules in the update operator.**
>
> We do not claim the idea of an iterative update mechanism nor the differentiable bundle adjustment to be contributions of our method. Please see above.
>
> **It is not theoretically sound to support the claim that the patch-based approach improves the accuracy of pose estimation results over the dense flow based approach when the patches are selected randomly rather than feature based methods.**
>
> Our empirical results support the claim that DPVO, which uses randomly selected patches, outperforms DROID-VO, which uses dense flow (see tables [1,2,3,4]). This result may be counter-intuitive, but it is not a weakness of our approach.
>
> We believe one possible theoretical reason is that random patches reduce redundant computation, and allow us to effectively re-allocate the computation and network capacity to learn better features for matching. We will revise our paper to clarify this point.
>
> **When it comes to real datasets, the results are mixed.**
>
> On real datasets, we outperform other methods in terms of *average error”, which is a standard way to compare performance.  The results are “mixed” only in the sense that we are not No. 1 on every single test sequence, but requiring superiority on every single test example would be an unusually strict standard.
>
> **The selection of baseline methods is inconsistent and questionable in Table 1-4, the ORB-SLAM version used is different.**
>
> ORB-SLAM has multiple versions, and on each dataset we use the version that produces the best results.
>
> The appearance of inconsistency, as mentioned in the caption of table 4, is because  ORB-SLAM3 would not work on TUM-RGBD despite our best efforts; The v1.0 implementation (the latest) fails to produce any poses for the keyframes.
>
> **authors did not compare the results to ORB-SLAM or D3VO on the EuRoC MAV dataset.**
>
> We omitted ORB-SLAM on the EuRoC table because we follow standard practice of VO evaluation [e.g. TartanVO ‘21] which focuses on comparing  strictly VO methods, i.e., those that do not perform global optimization, as opposed to SLAM methods that include global optimization. Global optimization can substantially improve results on input sequences with loops (such as those on EuRoC). As a result, VO methods are typically not expected to outperform SLAM methods and are not evaluated with SLAM methods as baselines.
>
> We omitted D3VO as it would  not be a fair comparison - D3VO performs unsupervised training on 5 of our 11 test sequences , and then evaluates on the remaining ones which contain the same scenes.
>
> This is a non-standard setting and gives D3VO an unfair advantage because unsupervised training on the same test scenes means that a method can essentially perform offline 3D reconstruction in advance for the same scenes and memorize the solutions. In offline reconstruction, a system can access all frames, including those from the future, whereas a VO system cannot use future frames. In contrast, our method is trained on synthetic data only, and is evaluated on each test sequence zero-shot, on the fly, without prior exposure to the same scenes.
>
> With the above caveats, in Table A in the rebuttal PDF, we nonetheless report results of D3VO and ORB-SLAM3 on EuRoC. We observe that D3VO outperforms DPVO on a subset of the evaluation sequences. This is expected because D3VO performs test-time-training on the remaining sequences that contain the same scenes, effectively allowing offline 3D reconstruction of the same scenes in advance. In contrast, DPVO is evaluated on all the EuRoC test sequences zero-shot, trained only on synthetic data. ORB-SLAM3 outperforms DPVO on all sequences (except it catastrophically fails on V202), but this is to be expected because it uses global optimization.
>
> **There are better results can be found for running ORB-SLAM on TUM-RGBD fr1 and TartanAir sequences (reported in https://paperswithcode.com/paper/droid-slam-deep-visual-slam-for-monocular/review/).**
>
> It is not clear that there are better results. ORB-SLAM has three versions (ORB-SLAM1, ORB-SLAM2, & ORB-SLAM3), and the cited URL reports results from ORB-SLAM1. We reported ORB-SLAM3 results because it is newer and has fewer catastrophic failures (0/16 versus 2/8) than ORB-SLAM1. Considering  the catastrophic failures, it is not clear which version is better.
>
> Regardless, we include both versions in Table G in the rebuttal PDF and show that this choice has no impact on our conclusions. Our approach has better results than both versions of ORB-SLAM on TUM-RGBD and TartanAir.
>
> Specifically, we outperform or match ORB-SLAM 1 & 3 on all TartanAir sequences.
>
> Both DPVO and DROID-SLAM report identical results for running ORB-SLAM3 on TUM-RGBD:  [X 0.017 0.210 X 0.034 X X X 0.009]
>
> DROID-SLAM also reports ORB-SLAM2 results on TUM-RGBD, which are even worse than ORB-SLAM3 (it fails on more sequences): X 0.071 X 0.023 X X X X 0.010
>
> **Response to Q1**
>
> Yes, we provide a discussion of this exact question in section G of the supplement. Tldr; we suspect that random patches have the most uniform coverage out of the 5 tested methods, which may help robustness.
>
> **Response to Q2**
>
> Addressed above.
>
> **Response to Q4**
>
> The running parameters are fixed across all datasets in the default and fast variants, respectively.
>
> **Limitation: The model can only be trained on synthetic datasets with ground truth optical flow.**
>
> This is not true, our method is not limited to synthetic datasets; it can be trained on any video dataset with poses and depth (e.g. Scannet, SUN3D, MannequinChallenge, among many others).
>
> As stated on L85-86 of the supplement, we only train on a synthetic dataset to fairly compare with DROID-SLAM.
>
> <ran out of space>

---

> > ### Comment · Reviewer_Uwmu · 2023-08-18
> >
> > Thanks for your rebuttal, I am satisfied with the newly added experiment results. They are very comprehensive. However, I still wish to address the following concerns:
> >
> > 1. The novelty: theoretical contribution of this paper is limited, whereas the sparse patch alignment is already used in other sparse direct methods, such as SVO, and neural iterative updates plus differentiable bundle adjustment is the design from DROID-SLAM. The system is undeniably valuable from the perspective of engineering, which brings some new insights into integrating traditional VO methods with deep neural networks.
> >
> > 2. The authors mentioned that it is affordable for DROID-SLAM to make more mistakes with dense flow fields in the rebuttal message, which means DPVO is prone to incorrect matches and is potentially less robust than DROID-SLAM. In order to compensate for the less redundancy, they used a larger spatial resolution and optimization window than DROID-SLAM. To be more specific, are there any differences in spatial resolution and optimization window size between DROID-SLAM and DPVO in the experiments? Would it be unfair to compare their accuracy under different optimization window sizes?

---

> > > ### Author Response · Authors · 2023-08-19
> > >
> > > **On novelty:**
> > >
> > > Thank you for your comment. Regarding novelty, although sparse matching in itself was not new, it was not obvious whether it would work at all in the framework of DROID-SLAM, because the SOTA performance of DROID-SLAM was understood to be dependent on the dense flow. While it may look simple and obvious in hindsight, making sparse matching work in DROID-SLAM was far from straightforward, for reasons we detailed in the global message.
> > >
> > > **On fair comparisons between DROID-SLAM (frontend) and DPVO:**
> > >
> > > DPVO does use larger optimization windows and higher spatial resolution than DROID-SLAM, but these are not unfair advantages, because they would be prohibitively expensive in DROID-SLAM, but are feasible in DPVO precisely because of the novel design with sparse matches.
> > >
> > > High GPU memory cost is already a major drawback of DROID-SLAM. Increasing its optimization window and resolution to that of DPVO would increase its already high memory cost by additional factors of ~14x and ~4x, respectively. This would require ~476GB of GPU memory and make DROID-SLAM practically useless.
> > >
> > > The main contribution of DPVO over DROID-SLAM is the efficiency improvement. Thus it is fair to compare efficiency at the same level of accuracy, allowing each system to pick its own hyperparameters appropriate for its design. Thus it is reasonable to use the default settings of DROID-SLAM, and pick the appropriate window size and resolution for DPVO to match the accuracy of DROID-SLAM and then compare efficiency. This comparison is valid and fair for the purpose of supporting our claim that DPVO has better efficiency than DROID-SLAM (frontend).
> > >
> > > Note that the estimated memory cost of increasing the optimization window size/density is because of the increase in the number of unique pairs of connected frames (383 in DPVO on average, 28 in DROID-VO on average. Both measured on TUM-RGBD.)

---

> > > > ### Comment · Reviewer_Uwmu · 2023-08-21
> > > >
> > > > Thanks for the response. The rebuttal has addressed my concerns. Thus, I increase my rating. However, I agree with other reviewers on the limited methodology novelty.

---

### Author Rebuttal · Authors · 2023-08-10

**Message to all reviewers regarding the novelty of DPVO:**

Our novelty is sparse matches *integrated with* neural iterative updates and differentiable bundle adjustment, which has not been done before and is nontrivial to design. While sparse matches have been used in prior work, they were not integrated with neural iterative updates and differentiable bundle adjustment.

Our method can be understood as a sparsification of DROID-SLAM, a state-of-the-art SLAM approach, to improve efficiency, but such sparsification is nontrivial. DROID-SLAM does iterations through recurrent 2D convolutions on 2D feature maps, producing a dense flow field between existing and incoming video frames. Naive sparsification schemes would encounter the following difficulties:

-  Simply sparsifying the feature maps and applying 2D convolutions does not work because we would be doing 2D convolutions on a feature map that is 95% empty

-  Simply replacing K x K convolution filters with 1x1s means losing the ability to use neighboring spatial context when making predictions - a fundamental benefit of 2D convolutions.

-  Using 95-99% fewer keypoint matches means there is less redundancy against incorrect matches. DROID-SLAM can afford to make many more mistakes since it produces a dense flow field rather than sparse matches.

Our solutions to these above challenges are:

-  Incorporating contextual information using patch-based correlation, and message passing between patches.

-  Adding 1D convolutions along the temporal dimension of a trajectory to pass around additional contextual information.

-  Offset the potential accuracy loss from the less redundancy  by using the significant memory savings to use a higher spatial resolution (¼ instead of ⅛) and use a larger optimization window.


Another contribution of this paper is a VO system that is open-source,  runs at 60-120 FPS, uses little GPU memory, and is as accurate or better than the current state-of-the-art open-source VO system (which is 2x-3x slower, 2x as expensive).
We believe a well-engineered open-source VO system is valuable and of interest to the research community.

---

### Decision · Program_Chairs · 2023-09-21

**Decision:**

Accept (poster)

**Comment:**

This submission received mixed reviews (WA, BA, BA, R). There was a consensus among the reviewers that the technical contribution is limited, even after rebuttal. However, the contribution is sufficiently interesting for most of the reviewers to propose to accept the submission. After checking the contribution (replacing dense matching by sparse matching in "neural visual odometer" because sparse matching is more reliable and accurate) carefully, I agree it is interesting, and I propose to accept this submission.